A novel privacy protection method of residents’ travel trajectories based on federated blockchain and InterPlanetary file systems in smart cities

Liu Fenghan
Wang Pan wangpan@njupt.edu.cn
School of Modern Posts, Nanjing University of Posts and Telecommunications , Nanjing, Jiangsu , China
Akleylek Sedat
Electronic publication date: 2023 Jul 27
Publication date: 2023
Volume: 9
Electronic Location ID: e1495
Received 2022 Dec 29; Accepted 2023 Jun 27
Copyright: © 2023 Liu and Wang
Copyright year: 2023
Copyright holder: Liu and Wang
License: This is an open access article distributed under the terms of the Creative Commons Attribution License, which permits unrestricted use, distribution, reproduction and adaptation in any medium and for any purpose provided that it is properly attributed. For attribution, the original author(s), title, publication source (PeerJ Computer Science) and either DOI or URL of the article must be cited.
License URL: https://creativecommons.org/licenses/by/4.0/

Keywords: Data sharing, Data privacy, Data security, Resident travel trajectory, Hyperledger fabric, IPFS, Blockchain

Funding: National Natural Science Foundation (General Program) 61972211 National Key Research and Development Project 2020YFB1804700 Future Network Innovation Research and Application Projects 2021FNA02006 and 2021 Jiangsu Postgraduate Research Innovation Plan KYCX210794 The article is sponsored by the National Natural Science Foundation (General Program) Grant 61972211, China, the National Key Research and Development Project Grant 2020YFB1804700, China, the Future Network Innovation Research and Application Projects No. 2021FNA02006 and the 2021 Jiangsu Postgraduate Research Innovation Plan under Grant No. KYCX210794. The funders had no role in study design, data collection and analysis, decision to publish, or preparation of the manuscript.

==============================
The government does have to record and analyze the travel trajectories of urban residents aiming to effectively control the epidemic during COVID-19. However, these privacy-related data are usually stored in centralized cloud databases, which are prone to be vulnerable to cyber attacks leading to personal trajectory information leakage. In this article, we proposed a novel secure sharing and storing method of personal travel trajectory data based on BC and InterPlanetary File System (IPFS). We adopt the Hyperledger Fabric, the representative of Federated BC framework, combined with the IPFS storage to form a novel mode of querying on-chain and storing off-chain aiming to both achieve the effectiveness of data processing and protect personal privacy-related information. This method firstly solves the efficiency problem of traditional public BC and ensures the security of stored data by storing the ciphertext of complete personal travel trajectory data in decentralized IPFS storage. Secondly, considering the huge amount of information of residents’ travel trajectories, the method proposed in this article can obtain the complete information under the chain stored in IPFS by querying the index on the chain, which significantly improves the data processing efficiency of residents’ travel trajectories and thus promotes the effective control of the new crown pneumonia epidemic. Finally, the feasibility of the proposed solution is verified through performance evaluation and security analysis.

Introduction

Coronavirus disease 2019 (COVID-19) is an infectious disease that is caused by severe acute respiratory syndrome coronavirus 2 (SARS-CoV-2) (Coronaviridae Study Group of the International Committee on Taxonomy of Viruses, 2020). The disease has spread into most nations across the globe, and there is still no specific drug for COVID-19. Non-pharmacological interventions (NPI), such as obtaining the travel trajectories of urban residents and regularly quarantining people who have visited outbreak areas, have been effective in reducing public exposure and thus slowing the spread of the disease (Bootsma & Ferguson, 2007), such as the travel codes used by the Chinese government, which have successfully reduced the number of cases and prevented the mass spread of COVID-19. So for today’s city dwellers, trip codes (proof of personal trip trajectory) have become a daily travel pass.

However, the trip codes of urban residents contain a large amount of private personal information, including but not limited to identification information, and personal trip trajectories. These private data are usually stored in centralized databases with the following problems: (1) inability to address centralized flaws such as server vulnerability to single point of failure (SPoF) and DDOS attacks: When the central server is subject to external attacks such as malware or brute force cracking attempts, data are at risk of loss and leakage. There have been several incidents of cyber violence against patients due to the leakage of COVID-19 patient travel track information. (2) Lack of trusted partnerships: since resident travel trajectory data contains a lot of private information, various departments are reluctant to lose control of this high-value data leading to difficulties in sharing data between different organizations in a timely manner, thus greatly reducing the efficiency of cross-departmental collaborative epidemic prevention. (3) Transparency issues: The data management and control currently used usually lack transparency, and there is a possibility of misusing the data if a malicious user gets hold of the high-value resident travel trajectory data.

Blockchain, as a typical privacy computing method (Lu et al., 2022), offers the possibility to solve the above problems. Blockchain, as a decentralized distributed ledger, abandons third-party management, and data are stored on all nodes on the blockchain, and each node jointly supervises and maintains this distributed ledger, so the risk of data tampering and loss is almost non-existent. To democratize control of the blockchain, the blockchain provides decentralized, transparent databases that give all participants equal access to the stored data and seeks consensus from all participants before agreeing to any changes. This decentralized trust model provides a trustworthy partnership among the participants to be willing to share data.

According to different access permissions, blockchain can be divided into the public chain, private chain, and alliance chain (Dib et al., 2018). All nodes on the public chain are open to the public, and all people can read and write on the blockchain and get the complete blockchain data, and it is difficult to reach a consensus for these nodes that come and go at will. Therefore, public chains are limited by the consensus mechanism and are prone to low throughput rates, and are difficult to scale when applied to application scenarios with large amounts of data. For example, the Bitcoin system can process up to seven transactions per second and the Ether platform can process up to 25 transactions per second (Xie et al., 2019). A private chain, on the other hand, completely binds the operation authority to an organization, so that data can only be managed and traced within a single organization, which improves the speed of data circulation, but its closed nature hinders the sharing of data among multiple parties. A Federated chain is a blockchain managed by several organizations that jointly participate, each running one or more nodes, in which the data are only allowed to be read and sent by different organizations in the system and jointly record the transaction data. The nodes of the coalition chain have a high degree of trust, they can use a relatively loose consensus mechanism, so the data processing speed will be greatly improved than that of the public chain, solving the problem of low efficiency of the public chain, at the same time, the data can be shared among the associated nodes, solving the problem of poor data circulation of the private chain.

The amount of travel trajectory data generated by city residents every day is huge, and due to the limitation of block size in the blockchain and the characteristic that data can only be added but not deleted, the blockchain ledger will keep expanding, so it is unrealistic for the data to be completely on the chain. In response to this problem, many scholars have proposed a combination of blockchain and cloud storage to store the original data under the chain, thus relieving the storage pressure on the blockchain. Zhao et al. (2021) presents a proposed blockchain based duplicate data elimination system for cloud data discrepancy authorization, which optimizes the traditional proof-of-vote (PoV) consensus algorithm, simplifies the existing discrepancy authorization process, and enables trusted management and dynamic update of permissions. However, due to the lack of effective encryption algorithms, the confidentiality of the data cannot be guaranteed. El Ghazouani, El Kiram & Er-Rajy (2019) proposed a storage model based on blockchain and cloud for medical data management, which uses on-chain storage of core data and off-chain storage of real data to ensure the integrity and security of medical data and take access control to ensure the security of data, but it cannot achieve efficient sharing of data. Zheng et al. (2018) proposes a solution that combines blockchain, cloud storage, and machine learning to achieve medical data sharing, but its complete medical data is stored in the centralized servers of the third party under the chain, and there is still a risk of data leakage if the third party is threatened. An interplanetary file system (IPFS) is a decentralized, distributed storage system based on content addressing. After depositing data into the IPFS system, the system returns a hash value derived based on that data. According to the principle of the hash function, even if only slight changes are made to the information, a completely different hash value will be obtained. This feature will ensure that data on IPFS will not be tampered with, ensuring the security of the data under the chain. So Wu & Du (2019) proposed a model for the secure sharing of electronic medical records using blockchain combined with IPFS to store the complete electronic case data in the IPFS system, but the model uses data masking to partially generalize the electronic case privacy information that needs to be deposited into the IPFS system, which is not secure enough. We need a more secure method to protect the privacy of urban residents’ travel trajectories.

Based on the above analysis, the goal of this article is to protect the security of urban residents’ travel trajectory data while achieving efficient data sharing among departments fighting against the epidemic, so a new approach to protect the privacy of smart urban residents’ travel trajectory based on federated blockchain and the interstellar file system is proposed. The key contributions are summarized as follows: 1) Designed a Hyperledger Fabric federated chain framework combined with an IPFS distributed storage model for data sharing while ensuring the security of city residents’ travel trajectories. Hyperledger Fabric is a pre-specified party-based permissioned blockchain that allows data sharing without relying on a single permission, while selective disclosure of data can better protect the privacy of sensitive data such as residents’ trip trajectories. Secondly, the use of IPFS to store the complete resident travel trajectory data reduces the storage pressure in the blockchain system and ensures the security of the data under the chain.

2) We propose an on-chain lookup and off-chain storage model, in which we store the complete AES-encrypted resident travel traces in IPFS, combine the returned IPFS storage address, AES key, Location_ID, and file’s creation time to form an index, and then store the index on the blockchain. Users can query the required data by Location_ID and file’s creation time. In this way, it is possible to ensure that personal privacy information will not be leaked and to search all the travel data sent and received and the corresponding transaction details through the index in the massive travel track data, thus improving the efficiency of epidemic prevention and control.

3) The security of our proposed approach is analyzed in detail and the performance of our approach (including data processing throughput and latency) is evaluated using the performance testing tool, Hyperledger Caliper.

Related work

The trajectory data of urban residents can usually reflect the living habits of residents, and there have been related research works that if an attacker analyzes the user’s trajectory data, they can not only obtain the user’s home address, workplace, and hobbies, but also further infer the user’s interest preferences, travel patterns, social habits, and other private information, and the attacker can even use the spatial and temporal correlation in the user’s location samples to infer other location information on the user’s trajectory (De Montjoye et al., 2015). In the context of the uncontrolled spread of COVID-19, the normal life and work of urban residents have been greatly affected by it. The prevention and control department has taken effective measures to prevent the further spread of COVID-19 by acquiring the travel trajectories of urban residents, while it is important to protect the privacy and security of their travel trajectory data.

Blockchain-based privacy protection for data in smart cities

Blockchain technology can provide data immutability to ensure the authenticity and accuracy of data in smart city management. Considering that users in smart cities require a wide range of services on a daily basis, which leads to a large amount of user personal information being collected and controlled by service providers, Mohammadinejad & Mohammadhoseini (2020) introduces an automatic access control manager based on blockchain networks without the presence of a third party, creating a secure channel between users and services and the possibility of their databases being misused or hacked. She et al. (2019) proposed a homomorphic federated chain for SHS sensitive data privacy protection (HCB-SDPP) based on traditional smart home systems. The model designs a homomorphic encryption algorithm based on pallier encryption to encrypt smart home data, and uploads sensitive data from encrypted all gateway nodes to the federated chain to ensure the security of sensitive data. The convergence of healthcare and smart cities can enable the use of health and medical data from around the world, which also poses challenges to the privacy of patients’ health information and the security of nearby mobile health users. Tripathi, Ahad & Paiva (2019) proposes a blockchain-based smart healthcare system framework to provide the inherent security and integrity of the system and ensure that the privacy of patient data is not compromised, Zhang & Lin (2018) proposed a PHI (private health insurance) data privacy protection scheme combining federated and private chains. In this scheme private blockchain is used to store the PHI of hospitals and the federated blockchain is responsible for recording the secure index of PHI and using public key encryption and keyword search to achieve data security and privacy protection on the federated blockchain. The Internet of Things (IoT), as a gas pedal for smart cities, requires the extensive use of distributed smart devices to collect and process data in smart city infrastructures. Kumar et al. (2021) proposes a specialized protection and security framework (PPSF) based on blockchain, which supports Internet-driven smart cities through two key mechanisms: privacy protection mechanism and intrusion detection mechanism.

IPFS in data privacy protection

IPFS, with its high security and efficiency, has also been heavily used in recent years in the area of data privacy protection. Liang et al. (2022) and Song et al. (2022) use an improved homomorphic encryption mechanism and SM4 homegrown commercial security algorithm to encrypt raw data, proposing a federal blockchain-based privacy protection scheme for personal data. The federated blockchain combined with IPFS to store encrypted data improves data transfer efficiency, safeguards user privacy and security, and proposes ciphertext policy attribute-based encryption to achieve fine-grained access control. In the IoT research, Hasan et al. (2022) proposes an advanced security model for IoT multimedia data sharing through blockchain combined with IPFS, which solves the security issues brought by centralized data storage. In the IoT research, Dhar, Khare & Singh (2022) proposes an advanced security model for IoT multimedia data sharing through blockchain combined with IPFS, which solves the security issues brought by centralized data storage. Kumar & Tripathi (2021) proposes a federated blockchain network with smart contract support to ensure security and privacy in the Internet of Medical Things (IoMT). The network integrates Interplanetary File System (IPFS) cluster nodes to secure and authenticate devices and provide secure storage management through IPFS cluster nodes. Gupta, Shukla & Tanwar (2020) proposed an approach called AaYusH (Ethernet Smart Contract (ESC) and IPFS-based TS system) to provide technical support for telesurgery systems, in which IPFS effectively reduces storage costs and provides reliable privacy protection of system data. In order to alleviate the problems of limited scalability and poor storage scalability brought by blockchain, which lead to the inability to directly combine blockchain with IoT under existing conditions. Li, Yu & Wang (2022) proposes a three-tier architecture blockchain (TBchain) architecture combined with IPFS to provide high performance privacy protection for IoT data stored on the blockchain. In the Dwivedi, Amin & Vollala (2021), IPFS is combined with blockchain to achieve car insurance driving data sharing for traffic congestion mitigation by applying it to a vehicle-based self-organizing network (VANET). In this scheme, the decentralized characteristics of blockchain and IPFS are used to achieve fully distributed storage of data and ensure the security of vehicle driving data. Focusing on EHR privacy protection in healthcare management, (Jayabalan & Jeyanthi, 2022) integrates the blockchain framework with the Interplanetary File System (IPFS) to enable healthcare organizations to maintain fail-safe and tamper-proof healthcare ledgers in a decentralized manner. Meanwhile, Lin & Zhang (2021) proposes a privacy protection method based on blockchain and improved IPFS to alleviate data privacy issues in IPFS by adding a data access mechanism to prevent potential privacy breaches during IPFS transmission.

Inspired by the above-related work, this article proposes a privacy protection method for smart city residents’ travel trajectories based on federated blockchain and interstellar file system, which assists government departments in effectively controlling the spread of epidemics while ensuring the security of city residents’ trajectory data.

Preliminaries

Residents’ travel trajectories data

A resident trajectory is a collection of data formed by a resident visiting a location at a certain time. When a resident visits a location at a certain time, the outbreak prevention and control department can obtain the resident’s geographic location in real-time through official location service providers including but not limited to GNSS, Bluetooth, cell towers, and WiFi.

As shown in Fig. 1, ( l1, t1) means that the resident visited location l1 at time t1, after which all the collected location information of the resident in 1 day is integrated to form the travel track data of the resident on this day. The complete resident travel trajectory data includes information such as personal name, cell phone number, file’s creation time, names of all visited locations, and specific geographic locations of visited locations. For the convenience of discussion, the travel trajectory data is simplified into three parts: user ID, visit location and visit time in this article. For example, the travel trajectories of residents U1, U2, and U3 for 1 day are shown in Fig. 1, then the travel trajectory of U1 for that day is expressed as U1=⟨(l1,t1),(l3,t2),(l2,t3),(l4,t4)⟩, the travel trajectory of U2 is expressed as U2=⟨(l1,t1),(l5,t2),(l6,t3)⟩, and U3=⟨(l4,t1),(l2,t2),(l3,t3),(l6,t4)⟩. The local epidemic prevention and control department collects the travel trajectories of all residents living in the area in 1 day, forming the travel trajectory data RTT = { U1, U2, U3} of the residents in the area on this day.

Figure 1 Residents travel trajectories.

Hyperledger Fabric

Hyperledger Fabric is a distributed ledger solution platform based on a modular architecture that provides a high degree of confidentiality, resilience, flexibility and scalability. Hyperledger Fabric is a non-fully public blockchain that helps protect the privacy of residential travel data, which is often highly sensitive. Hyperledger Fabric does not require expensive mining calculations to submit transactions through pre-specified membership and strict access policies, so it helps build blockchains that can scale with shorter latency of the blockchain. To ensure the security of communication, Hyperledger Fabric supports TLS for transport layer security for communication between nodes, and the ledger is only shared among organizations participating in the common bookkeeping, while other organizations do not have access to the ledger, setting strict permission control for accessing resources. In terms of privacy protection Hyperledger Fabric implements a private data mechanism to solve the problem of allowing certain private data to be shared among only a small number of organizations in a ledger with multiple participating organizations.

Interplanetary file system (IPFS)

IPFS is a distributed file storage system that assigns unique storage addresses to files uploaded to the system and the hash addresses of IPFS files cannot be changed, so IPFS is a tamper-proof storage mechanism. It differs from the HTTP protocol in that the data is stored in a distributed manner in the system built on IPFS, whereas the HTTP protocol places the data on the server side, which puts tremendous pressure on the server. IPFS addressing is based on the principle of content addressing, and when data is stored in the IPFS system, the system returns a hash value based on that data. According to the principle of the hash function, even if only minor changes are made to the information, a completely different hash value will be obtained. This feature will ensure that data on IPFS will not be tampered with. When requesting data from the IPFS file system, it will use the distributed hash table to find the node where the requested data is located and request the data back in that node. these features of IPFS make the perfect combination of blockchain technology at the application level to provide a distributed storage solution for traditional blockchain programs.

Methods

In this section, we will first explain the framework of our proposed method and related entities and terms, and then describe in detail the concrete implementation process of the privacy protection method of smart city residents’ travel trajectories based on federated blockchain and interstellar file system.

Entities and nouns

The system model framework of the proposed approach in this article is shown in Fig. 2, and the entities and terms involved in the system framework diagram are explained next in the following: 1) PKI/CA: Public key infrastructure (PKI)/certification authority (CA) is a trusted third party (in this system, it is a government department) that generates a CID corresponding to the identity of each user who joins the system. Based on the CID, the attribute authority sends the identity-related attribute set and key to the user.

2) Departments: Departments are the epidemic prevention and control departments in different regions. The epidemic prevention and control departments in each region collect the travel data of residents through the base stations and WIFI connected to the residents’ cell phone signals and share the trajectory data with the epidemic prevention and control departments in other regions in the system.

3) Users: Users in this system refer to the staff of the epidemic prevention and control department, who is responsible for uploading, querying, and downloading the specific operations of the residents’ travel trajectory data.

4) Block_Index: To relieve the storage pressure of the blockchain and ensure the security of the travel track data, the system only stores the easily searchable Block_Index on the blockchain. Block_Index consists of the storage address of residents’ travel trajectory in IPFS, AES encryption key, Location_ID, and CreateTime. The storage address of IPFS and the AES key is encrypted by the public key of the data recipient, and the geographic location of the resident’s travel track data and the CreateTime are not encrypted for the convenience of the query.

5) IPFS: Due to the practical limitations of cost, storage capacity, and other factors, a large amount of residents’ travel track data is stored encrypted outside the blockchain. IPFS can assist the blockchain to store the cipher text of residents’ travel track uploaded from the client by the staff of the epidemic prevention and control department to reduce the storage pressure of the blockchain.

6) Federated Blockchain: Blockchain can be considered a secure distributed database ledger. The federated blockchain network in this system consists of different regional epidemic control authorities, which store the encrypted index on the blockchain to ensure security.

Figure 2 System architecture.

Method implementation

In general, our method uses AES encryption for residents’ travel trajectory data in advance, uploads the encrypted data to IPFS network and obtains the corresponding storage address in order to relieve the storage pressure of blockchain, and stores the address and AES key, Location_ID and file creation time to form an index to the federated blockchain network, through which users can query the data. At the same time, through the private data mechanism of the federated blockchain, it achieves that the travel trajectory data is only shared among a few organizations in the ledger of the participating organizations to exclude the risk of privacy leakage. The proposed approach is described in detail in the following four aspects: data encryption, data upload, data sharing, and data query. To simplify the description, the meaning of some special characters are shown as Table 1.

Table 1 Notations description.

Notations	Descriptions	
RTT	Residents travel trajectories	
CRTT	Encrypting residents’ travel trajectory data	
K	AES key	
AESb, AESd	AES encryption function	
PKR, PKS	Receiver’s public key and sender’s public key	
SKR, SKS	Receiver’s private key and sender’s private key	
HashCRTT	Hash address of resident travel trajectory data stored in IPFS	

Staff U of the epidemic prevention and control department sends an application to the system, the PKI/CA generates an ID corresponding to its identity, and according to the ID, the attribute authorization authority sends the identity-related attribute set S and the public key PKid and private key SKid to the staff.

As shown in step 1 of Fig. 2, the local epidemic prevention and control department cooperates with official operators to obtain the location information of residents in real-time through the base stations accessed by residents’ cell phone signals and connected WIFI and integrates them to obtain the complete travel trajectory data RTT of all residents in the area (a detailed description of travel trajectory data can be found in Fig. 1).

AES encryption

AES encryption algorithm is a type of symmetric encryption algorithm, which has the advantages of high security, high efficiency and resistance to traditional password cracking. To ensure that no privacy leakage occurs during the uploading of residents’ travel track data to IPFS, the method proposed in this article preemptively encrypts the travel track data to be uploaded to IPFS with AES encryption. Assuming that the staff of an epidemic prevention and control department in a region uses AES encryption function as AESb, the encryption generates encrypted travel track data CRTT:

(1) CRTT=AESb(K,RTT)

Identity verification

Algorithm 1 shows the authentication of the epidemic prevention and control authorities in each region of the federated network for the upload and sharing of resident travel track data. The purpose of this algorithm is to prevent malicious nodes from accessing the federated network and ensures the security of the proposed model. By joining the federated blockchain network, they receive a unique proof of identity, known as a Proof of Identity (PoI). In the Fabric network, the user provides a CA-generated certificate representing his identity and the MSP determines whether the user is trustworthy by identifying him or issuing him with a CA for his identity.

Algorithm 1 Algorithm for peer nodes verification in consortium network.

  Input: input Proof-of-Identity (PoI)	
  Output: output verification of peer nodes	
1 if msg.sender is not valid then	
2    Return Invalid peer node;	
3  else	
4    Peer node can upload/share/query/download RTT;	
5    Return Valid peer node;	
6  end	

Data upload

The process of uploading residents’ travel track data is shown in steps 3, 4, and 5 of Fig. 2, RTT is pre-encrypted with AES algorithm to form CRTT, and the staff US who uploads the data uses their private key SKS to generate the signature SigSKS (CRTT), and then sends the ciphertext CRTT, POI, and signature SigSKS (CRTT) together to the IPFS node. Algorithm 2 describes the storage mode in which user data is uploaded to the system to form an on-chain storage index and complete data is stored under the chain; after receiving the ciphertext and signature uploaded by the user, the IPFS node verifies the identity of the user. After confirming the identity, the CRTT is uploaded to the IPFS network for storage. At the same time, the IPFS returns the unique hash address HashCRTT based on the uploaded ciphertext. Each node in the IPFS network uses the public key to sign the received hash address and sends CRTT, HashCRTT, and SigSKS (CRTT) to each IPFS node in the cluster for verification. Each IPFS node in the cluster verifies the user’s identity and calculates the CRTT hash locally. If it matches the hash value returned by the IPFS network, a confirmation message is sent to the IPFS node interacting with the user, and the complete resident travel track ciphertext is stored in the IPFS. The stored address HashCRTT returned by IPFS, AES key K, Location ID where the resident travel track data is located, and file’s creation time are formed into an index and encrypted using the public key PKR of the data recipient i.e., Enc ( HashCRTT, K, PKR) to form an encrypted index Block_Index:

Algorithm 2 Data upload.

  Input: input CRTT,Proof-of-Identity(PoI),K,location_ID,CreateTime	
  Output: output Algorithm result	
1 if Peer node is valid and Each IPFS node in the cluster verifies the identity successfully then	
2    CRTT Upload to IPFS;	
3    Return HashCRTT;	
4    if Consensus nodes reach consensus then	
5       Block_Index Upload to Blockchain;	
6       Return AddressIndex;	
7     else	
8       Return No consensus;	
9     end	
10 else	
11    Return Invalid identity;	
12 end	

(2) Block_Index=<Enc(HashCRTT,K,PKR),Location_ID,CreateTime>

Then Block_Index is sent to the federated blockchain network, the Fabric node receives the relevant information and sends it to other Fabric nodes to jointly verify its identity and content, deploys the transaction after successful verification, and sends it to each node in the blockchain federation in the node group, and the transaction is packaged on the chain after each node reaches consensus and the Block_Index is stored on the federated blockchain.

Data sharing

The proposed method uses physical deployment as shown in Fig. 3 to share data among each other. The figure mainly includes Fabric nodes, IPFS nodes, WEB Service, and PostgreSQL database for system functions and interactions, and each epidemic prevention and control department joining the network maintains two types of nodes, Fabric nodes, and IPFS nodes. In this article, the epidemic prevention and control department maintains the endorsement node and the orderer node in the Fabric node in the system. Epidemic prevention and control departments in the network store the index data of residents’ travel trajectories at the endorser node, the endorser node runs the smart contract and endorses the result of smart contract execution and stores the data, then the ordering node collects the data and packages the transactions that satisfy the endorsement strategy into blocks to be distributed to each epidemic prevention and control department in the blockchain network. The IPFS cluster consisting of IPFS nodes runs between the epidemic prevention and control departments to store the cipher text of residents’ travel track data after AES encryption to reduce the pressure of blockchain storage. Finally, the WEB server is responsible for communicating with the fabric nodes, IPFS nodes, and business database servers to connect all epidemic prevention and control departments, break the data silos, and realize the sharing of residents’ travel track data among all epidemic prevention and control departments, to achieve the purpose of collaborative prevention and control of the epidemic by all departments while effectively protecting the privacy and security of residents’ travel track data. Algorithm 3 is the specific implementation of data access in the data-sharing process, the user sends a request to access the data, and the user first authenticates through the POI, and the corresponding Block_Index of the CRTT to be accessed to verify whether the correct trajectory data is being accessed. If the above verification is passed, the user decrypts Block_Index with his private key SKR and obtains the storage address, AES encryption key, Location_ID, and CreateTime returned by IPFS. Through the returned IPFS storage address, the IPFS network is accessed, and the nodes in the IPFS node group first verify the user’s identity and then retrieve the complete resident travel track encrypted data at the corresponding storage location in the system’s private IPFS network to send to the user after the verification is passed. After receiving the encrypted resident travel track data, the user decrypts the cipher text by using the AES key obtained when decrypting the Block_Index. Assuming that the AES decryption function used by the staff of a regional epidemic prevention and control department is AESd and has AES key K, the user obtains RTT by decrypting.

Figure 3 Physical deployment architecture.

Algorithm 3 Data access.

  Input: input Proof-of-Identity(PoI),Block_Index,SKR	
  Output: output Algorithm result	
1 if PoI is valid and SKR is right SK then	
2    Get HashCRTT,K from Block_Index;	
3    M = HashCRTT;	
4    N = Get Hash Value from IPFS;	
5    if (M == N) and K is right AES_Key then	
6       Return RTT;	
7     else	
8       Return Data access failed;	
9     end	
10 else	
11    Return Access failed;	
12 end	

(3) RTT=AESd(K,CRTT).

RTT is the travel track data of the residents in that geographical location, and the user can download the data for use.

Data query

The specific process of data encryption, index construction, and data query by users is shown in Fig. 4. When the user sends a data query request, the user’s identity is first verified through the POI. Algorithm 4 is the query process of the encrypted index of the residents’ travel trajectory data. After the user passes the authentication, he enters his private key SKq, and the system returns the encrypted index of all the data related to the user through the smart contract deployed for query within the system, which includes the CRTTs of encrypted residents’ travel trajectory data sent by the user to other epidemic prevention and control departments and the CRTTR of encrypted residents’ travel trajectory data received by the user. First, the user receives the Block_Index of these data returned by the system, and the user queries the Block_Index of the travel trajectory data he wants to get by Location_ID and CreateTime, and if the query is successful, the user gets the data index Block_Indexquerry. The user decrypts Block_Indexquerry by SKq to obtain the storage address, AES encryption key, geographic location of resident travel track data, and CreateTime returned by IPFS, and finally obtain the complete residents’ travel track data by the Algorithm 3.

Figure 4 Processes of file encryption, index construction and search.

Algorithm 4 Block_Index querry.

  Input: input Proof-of-Identity (PoI),Location_ID,CreateTime,SKq	
  Output: output Algorithm result	
1 if Peer node is Valid and SKq is right SK) then	
2    BOD = Get Block_Indexs of CRTTs and CRTTR;	
3    if Location_ID and CreateTime are in the BOD then	
4       Return Block_Indexquerry;	
5     else	
6       Return Query failed;	
7     end	
8  else	
9    Return Access failed;	
10 end	

Results and analysis

Security analysis

Our proposed method ensures the integrity and privacy of residents’ travel trajectory data.

Integrity

The proposed approach in this article uses a combined blockchain and IPFS storage structure, where the user integrates Block_Index into the blockchain transactions and stores CRTT within IPFS. As shown in Fig. 5 either changes to Block_Index or CRTT will result in a change in the hash value, thus ensuring that the stored data is immutable and traceable. If an illegal user wants to modify the hash of the current data stored in the blockchain system at the same time, it must mimic the master chain like the source chain so that the corresponding transaction can be accepted by most nodes, which is almost impossible to achieve due to the limitation of computing power. According to the hash rule, the hash values obtained by hashing two different files are different; therefore, the forged CRTT and Block_Index will not be shared and accessible. For tampering attacks, the data stored in the blockchain is immutable unless the blockchain is threatened by a 51% attack.

Figure 5 Storage structure.

Privacy

The proposed approach in this article ensures the privacy of RTT from two aspects: 1. Pre-encrypted RTT using AES algorithm to form CRTT stored in IPFS, without the key K and HashCRTT is unable to access the CRTT. And the Block_Index containing K and HashCRTT is stored in the blockchain system, and the index cipher cannot be decrypted without the private key SKR of the data receiver. This ensures that only the designated data receiver can access the resident travel trajectory data sent out by the user2. As shown in Fig. 6, the Hyperledger Fabric federated chain architecture used in this article has a unique channel mechanism, and the data in different channels are isolated from each other and cannot be accessed arbitrarily, which means that all the data in Channel1 can only be accessed by Department1 and Department3. This enables the data sharing of residents’ travel trajectory data only among related epidemic prevention and control departments, ensuring the privacy of the data.

Figure 6 Fabric channel mechanism.

Performance analysis

Our system is deployed on a virtual machine with a 2-core CPU, 4GB RAM and Ubuntu 18.04 operating system, and we use the travel trajectory data of COVID-19 infected persons in Beijing, China in 2019, which is legally disclosed, as our test dataset. The performance testing consists of two main parts: (1) we first tested the upload/download time of files on IPFS and compared it with traditional cloud storage regarding the upload speed. (2) Querying and updating in smart contracts are the key operations of this system, and we used the Hyperledger caliper tool to test the performance performance of system queries and updates regarding the throughput rate and latency. The proposed method refers to the uploading of encrypted index of residential travel trajectory data and the query operation refers to the querying of encrypted index of residential travel trajectory data. We used Hyperledger caliper tool to test the performance of system queries and updates in terms of throughput and latency.

In the first experiment, we conducted an IPFS storage performance test, we selected files of sizes 1 MB, 5 MB, 10 MB, 50 MB, 100 MB, and 500 MB respectively, and tested the time required to upload files to IPFS and the time required to download files from IPFS, the test results are shown in Fig. 7A, from the figure we can find that the time required to upload files to IPFS is several times longer than the time required to download the file from IPFS. To prove that IPFS has a faster transfer speed, we compared the upload speed with traditional cloud storage. From Fig. 7B, we can see that when the file size is less than 10 MB, the upload speed of cloud storage is slightly faster than IPFS, but as the file size keeps getting larger, the upload speed of IPFS far exceeds that of cloud storage, and we have reason to believe that IPFS has better performance than cloud storage when transferring massive files.

Figure 7 IPFS performance testing.

Transaction latency is the average time elapsed from the time a transaction is submitted to the time it is added to the ledger and confirmed on the blockchain, and throughput is the number of valid transactions submitted per second (tps) on the blockchain in the second experiment, we fixed the number of transactions at 5,000 and obtained the average latency and throughput of the system by varying the send rate, and used this to analyze the performance of the system. We first tested the system query function for 10 rounds, increasing the send rate by 50 tps each time until 500 tps. Figure 8 shows the performance of the system query operation, from which the relationship between the performance of the chain code query function and the frequency of sending can be seen. From Figs. 8A and 8B, it can be seen that as the send rate value increases, both the throughput and the average latency increase. The system throughput reaches a maximum of 203 tps when the send rate reaches 300 tps, and the average delay increases with the send rate, and the delay increases significantly above 250 tps, indicating that the system can handle more than 250 tps, leading to transaction blocking.

Figure 8 Query operational performance.

Similarly, we test the system update function by increasing the send rate by 25 tps each time until 200 tps: Fig. 8 shows the performance of the system update operation. From Fig. 9A, we can see that as the send rate increases, the throughput rate also increases, and the system throughput reaches a maximum of 119 tps when the send rate reaches 150 tps, after which the system throughput decreases when the send rate increases further, and from Fig. 9B, we can see that the system average latency increases sharply when the Send Rate exceeds 125 tps. This situation indicates that the processing capacity of the fabric node reaches its maximum at around 125 tps.

Figure 9 Update operational performance.

Finally, we measure the performance metrics as the number of transactions to be processed by the blockchain increases from 200 to 2,000 tx. As shown in Fig. 10, we find that increasing the number of transactions leads to a linear increase in throughput, which starts to gradually decrease and flatten out when the number of transactions increases to 1,200 tx, which is the saturation point determined by the capacity of the nodes used in the test. In Fig. 11 the response time of the system query operation vs. the update operation is shown when the number of transactions goes from 200 to 2,000 tx, we can see that in the system the update operation takes more time than the query operation, at 2,000 tx it takes about 19s, but considering the performance of virtual machines we used for testing, we consider this to be within acceptable limits.

Figure 10 Throughput for different number of transactions.

Figure 11 Response time.

Discussion

In this section, we discuss existing approaches to the privacy protection of trajectory data and compare them with our proposed approach. The currently existing trajectory data privacy protection techniques are as follows: (1) Protecting trajectory data privacy based on suppression techniques: Chen et al. (2013) distinguishes the sensitivity of the trajectory data in the original trajectory dataset and selectively publishes it. That is, when releasing trajectory data, the locations that do not involve sensitive information are released directly, and the sensitive locations on the trajectory are not released or are released after certain anonymization processes are performed on them. (2) Based on the k-anonymity model to protect user trajectory data: Chow, Mokbel & Liu (2011) constructs the hidden region satisfying k-anonymity by peer-to-peer communication, and randomly selects a node in the anonymity region as the initiator of the query to send the service request, which achieves the effect of hiding the actual query node with the fuzzy region. (3) Blockchain-based trajectory privacy protection methods: Jeong et al. (2020) combined blockchain and cloud storage technology to achieve privacy protection of Telematics data including vehicle trajectory data and data sharing between different vehicles, and Zhu et al. (2021) proposed a scheme called B-PPLS based on Bitcoin system combined with OBE encryption algorithm to protect the privacy of user trajectory data and achieve data sharing, which guarantees the traceability of location data by blockchain the location data of different record owners in chronological order, and shares the location data among fully trusted requestors by trust degree division. A distributed framework combining IPFS and federated blockchain is proposed in Kumar & Tripathi (2020) for secure storage and sharing of reports including patient trajectory data.

As shown in Table 2, this article compares this article’s scheme with other existing trajectory privacy protection schemes based on four aspects: privacy protection, storage method, scalability, and shareability. These two types of trajectory data protection methods proposed in the Chen et al. (2013) and Chow, Mokbel & Liu (2011) only discussed the feasibility of the privacy protection aspect of trajectory data and did not discuss the shareability of data, system scalability, or data storage methods. In the previous article, we have discussed the shareability, scalability and the storage method of data of our proposed methods in detail. The scheme proposed in the Jeong et al. (2020) uses third-party cloud storage under the chain when the third party is threatened, the data still has the risk of leakage and cannot ensure the security of the data under the chain. The approach proposed in Zhu et al. (2021) is deployed on the Bitcoin system, which, as a representative of the public chain, sacrifices scalability to ensure complete decentralization. The method proposed in Kumar & Tripathi (2020) does not pre-encrypt the patient reports to be uploaded to IPFS, which can lead to the risk of data leakage during the uploading process. Regarding the problems in Jeong et al. (2020) and Zhu et al. (2021), the approach proposed in this article uses Hyperledger Fabric federated chain with good scalability and AES encryption technology to provide data privacy protection, and finally data storage through IPFS to relieve the storage pressure of the blockchain system and ensure the security of both on-chain and off-chain data.

Table 2 Comparison with other schemes.

	Privacy protection	Storage mode	Expandability	Sharing	
Chen and Chow	Yes	No discussion	No discussion	No discussion	
Jeong	No	Cloud storage+Blockchain	Yes	Yes	
Zhu	Yes	Blockchain	No	Yes	
Kumar	No	IPFS+Blockchain	Yes	Yes	
Our method	Yes	IPFS+Blockchain	Yes	Yes	

In this article, we have only proposed a rough solution to the problem, and many details are not yet perfect. For example, the contracts involved in this article all use serial transfers for transactions, and only the serial contracts are stress-tested in the experiments, but in reality, when large-scale data sending and receiving is required, parallel transfers are needed to obtain better throughput rates. And currently this article implements the simplest one-to-one data sending and receiving, which is not well supported for the more flexible one-to-many case. In the next section, we suggest some potential improvement directions for future research.

Conclusions

Resident trajectory data plays a huge role in epidemic prevention and control efforts, and its secure sharing will effectively inhibit the spread of COVID-19. In response to the current special demand for sharing resident travel trajectory data, this article provides a new approach for privacy protection of resident travel trajectories based on federated blockchain and interstellar file system. The proposed method uses the Hyperledger Fabric federated chain framework combined with the storage model of IPFS, which is used to store the cipher text of the complete travel trajectory data, and the blockchain keeps an encrypted index consisting of the AES key used to encrypt the resident travel trajectory data, the returned IPFS storage address, the Location_ID and the creation time of the file. Users can query the corresponding index according to the location_ID and the creation time of the file, and get the travel track data through the index. It protects the privacy and security of residents’ travel trajectory data while satisfying the demand of data sharing among various epidemic prevention departments.

In future research, we can improve the universality of the proposed method from the following perspectives. We can try to change the traditional blockchain transaction form into a parallel transaction form based on DAG to improve the throughput rate to adapt to large-scale data transmission, and DAG-based public chain projects such as IOTA and FTM have proved that they can bring more efficient and lighter transaction forms. Secondly, we can use some new blockchain consensus protocols, such as DPBFT protocol (Fu et al., 2021) and Dumbo protocol (Guo et al., 2020), to improve the fault tolerance rate of the system while ensuring the system performance.

Additional Information and Declarations

Competing Interests

Author Contributions

Data Availability

The authors declare that they have no competing interests.

Fenghan Liu conceived and designed the experiments, performed the experiments, analyzed the data, performed the computation work, prepared figures and/or tables, authored or reviewed drafts of the article, and approved the final draft.

Pan Wang conceived and designed the experiments, authored or reviewed drafts of the article, and approved the final draft.

The following information was supplied regarding data availability:

The raw data is available at GitHub and Zenodo: https://github.com/FenghanLiu/Raw-Data.

Fenghan Liu. (2023). FenghanLiu/Raw-Data: Raw Data (1.0). Zenodo. https://doi.org/10.5281/zenodo.7585056.

The code is available at GitHub and Zenodo: https://github.com/FenghanLiu/peerj-RTT.

Fenghan Liu. (2023). FenghanLiu/peerj-RTT: peerj-RTT (1.1). Zenodo. https://doi.org/10.5281/zenodo.7974584.

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
