# Peer review of "A novel privacy protection method of residents’ travel trajectories based on federated blockchain and InterPlanetary file systems in smart cities"

_PeerJ Computer Science, doi:10.7717/peerj-cs.1495_

## Round 0.1 · original submission · Minor Revisions

Please address the issues raised by the reviewers and prepare a revised manuscript.

Reviewer 1 ·

Basic reporting

This manuscript proposed a secure sharing and storing method for personal travel trajectory data adopting Block Chain and IPFS. Hyperledger Fabric was applied in the Federated BC framework, and IPFS was used to store user data. The English language in this work is professional except a few negligent mistakes. The Intro & Background, including citations, are well organized. The structure conforms to PeerJ standards. Raw data has been supplied. Some figures look unclear as follows:
(1) In Figure 2, the text needs to be redrawn in higher resolution.
(2) In Figure 4, the text ‘BlockChain’ needs to be redrawn in higher resolution.

Experimental design

The original primary research is within Scope of the journal. Research questions are well defined, relevant & meaningful. It is stated how the research fills an identified knowledge gap. Rigorous investigation performed to a technical & ethical standard. Methods described with sufficient detail &information to replicate.
As for the experimental design, some advice are as follows:
(1) If available, using public datasets about personal travel trajectory might be beneficial to the verification of this method during the experimental evaluation.
(2) Adding the comparison with current SOTA methods might be better.
(3) As for the Figure 6 IPFS Performance Testing, more files/data sizes are advised to be conducted in the experimental evaluation.

Validity of the findings

Impact and novelty not assessed. Meaningful replication encouraged where rationale & benefit to literature is clearly stated. All underlying data have been provided; they are robust, statistically sound, & controlled. Conclusions are well stated, linked to original research question & limited to supporting results.

Additional comments

no additional comments

Annotated reviews are not available for download in order to protect the identity of reviewers who chose to remain anonymous.

Reviewer 2 ·

Basic reporting

This manuscript proposed a secure sharing and storing method for personal travel trajectory data adopting Block Chain and IPFS. Hyperledger Fabric was applied in the Federated BC framework, and IPFS was used to store user data. The English language in this work is professional except a few negligent mistakes. The Intro & Background, including citations, are well organized. The structure conforms to PeerJ standards. Raw data has been supplied. Some figures look unclear as follows:
(1) In Figure 2, the text needs to be redrawn in higher resolution.
(2) In Figure 4, the text ‘BlockChain’ needs to be redrawn in higher resolution.

Experimental design

The original primary research is within Scope of the journal. Research questions are well defined, relevant & meaningful. It is stated how the research fills an identified knowledge gap. Rigorous investigation performed to a technical & ethical standard. Methods described with sufficient detail &information to replicate.
As for the experimental design, some advice are as follows:
(1) If available, using public datasets about personal travel trajectory might be beneficial to the verification of this method during the experimental evaluation.
(2) Adding the comparison with current SOTA methods might be better.
(3) As for the Figure 6 IPFS Performance Testing, more files/data sizes are advised to be conducted in the experimental evaluation.

Validity of the findings

Impact and novelty not assessed. Meaningful replication encouraged where rationale & benefit to literature is clearly stated. All underlying data have been provided; they are robust, statistically sound, & controlled. Conclusions are well stated, linked to original research question & limited to supporting results.

Reviewer 3 ·

Basic reporting

This manuscript proposed a secure sharing and storing method for personal travel trajectory data adopting Block Chain and IPFS. Hyperledger Fabric was applied in the Federated BC framework, and IPFS was used to store user data. The English language in this work is professional except a few negligent mistakes. The Intro & Background, including citations, are well organized. The structure conforms to PeerJ standards. Raw data has been supplied. Some figures look unclear as follows:
(1) The data query encryption index mentioned in the Method Implementation can be described in more detail, for example, by means of an accompanying figure.
(2) For the convenience of the reader, it is suggested that the nouns in METHODS be explained in a unified manner in a list.
(3) Figure 4 should provide a more detailed depiction of the storage structure and embellish the figure.

Experimental design

The original primary research is within Scope of the journal. Research questions are well defined, relevant & meaningful. It is stated how the research fills an identified knowledge gap. Rigorous investigation performed to a technical & ethical standard. Methods described with sufficient detail &information to replicate.
As for the experimental design, some advice are as follows:
(1) It is recommended to select larger files/data for IPFS performance testing in the Performance Analysis.
(2) Specific performance metrics in Performance Analysis, such as throughput rate, latency may be better compared to existing methods.

Validity of the findings

Impact and novelty not assessed. Meaningful replication encouraged where rationale & benefit to literature is clearly stated. All underlying data have been provided; they are robust, statistically sound, & controlled. Conclusions are well stated, linked to original research question & limited to supporting results.

---

## Round 0.2 · Minor Revisions

Please address the minor issues raised by the reviewers and resubmit.

Reviewer 3 ·

Basic reporting

This manuscript proposed a secure sharing and storing method for personal travel trajectory data adopting Block Chain and IPFS. Hyperledger Fabric was applied in the Federated BC framework, and IPFS was used to store user data. The English in this work is generally professional.The Intro & Background, including citations, are well organized. The structure conforms to PeerJ standards. Raw data has been supplied.
However, there are still some parts that are not clear enough, which affects the reader's reading experience, for example, lines 264-269 of Data Upload have unclear expressions, which needs to be checked again for the full English expressions.

Experimental design

The original primary research is within Scope of the journal. Research questions are well defined, relevant & meaningful. It is stated how the research fills an identified knowledge gap. Rigorous investigation performed to a technical & ethical standard. Methods described with sufficient detail &information to replicate.

Validity of the findings

Impact and novelty not assessed. Meaningful replication encouraged where rationale & benefit to literature is clearly stated. All underlying data have been provided; they are robust, statistically sound, & controlled. Conclusions are well stated, linked to original research question & limited to supporting results.

Additional comments

Comments on Discussion
The comparison of the proposed method with existing methods in the discussion section is not sufficiently elaborated, and a more detailed description of the advantages of the proposed method is recommended.

·

Basic reporting

1. Researchers propose a secure method for sharing and storing personal travel trajectory data during COVID-19 using blockchain and InterPlanetary File System (IPFS).
2. The method ensures data processing efficiency and protects personal privacy by storing ciphertext in decentralized IPFS storage and querying the index on the chain.

Experimental design

The experimental design involves using Hyperledger Fabric, a federated blockchain framework, combined with IPFS storage to achieve efficient data processing and protect personal privacy-related information. The method stores the ciphertext of complete personal travel trajectory data in decentralized IPFS storage, ensuring the security of stored data.

Validity of the findings

The feasibility of the proposed solution is verified through security analysis and performance evaluation.

Reviewer 5 ·

Basic reporting

For review “novel privacy protection method of residents travel trajectories based on federated blockchain and InterPlanetary file systems in smart cities” (#80680):

Read through this manuscript, I am impressed about their present research work. This manuscript does an excellent job demonstrating significant novel algorithms they have developed. In order to protect PII (Personally identifiable information) privacy, this is an excellent achievement with tracked record experiments result approval stated their research achievements in the subject areas. The last smart city research congress I participated does not have such a paper reported as detail as this manuscript reported yet.

In particular, I am very happy with the tables, equations and figures inside this paper, which have explicitly explained the details of their achievements. The experiment results and the analysis were excellently explained the significance of their invented algorithms contributions.

The paper could be both more compelling and useful to a broad readership if the authors could elaborate the PII privacy analytics in more details with the significant experiment results. Moreover, as the authors stated the limitation to date, a detailed recommendation of further development should be more impressive.

Nevertheless, in my opinion, a minor correction is necessary.

Details:

Looks like this paper was submitted in a rush, the whole paper needs a quick amendment. Although the citation was good and captured well, however, a few references have not been properly completed as the other reference in the paper’s references list. Every abbreviation should be explained when they first appears at the paper. There are still some flaws exist need to be corrected. The font of the titles of the tables and figures are not consistent. In addition, I think a numerical structure would demonstrate the authors’ logic even more clearer and properly. Furthermore, before its re-submission, a professional proofreading is suggested, in that way, it should ensure the publication quality of this very good paper as it deserves.

Overall, this is still a very good paper, with significant contributions to the subject area, but need properly amended in academic writing style, I think.

Experimental design

See above

Validity of the findings

See above

---

## Round 0.3 · Major Revisions

The referral process is now complete. While finding your paper interesting
and worthy of publication, the referees and I feel that more work could be
done before the paper is published. My decision is therefore to
provisionally accept your paper subject to major revisions. More details are needed.

Reviewer 6 ·

Basic reporting

In this paper, the authors comes with a statement about a new method of privacy protection based on federated blockchain and Interplanetary systems in Smart cities. The authors have addressed the reviewers' comments and concerns in their revised manuscript. The structure conforms to PeerJ standards. Raw data has been supplied.
But after reviewing this article, I have comments/suggestions that need to be addressed to improve the quality of the paper, these are:
- References are insufficient, and recent studies containing privacy protection methods based on the blockchain can be added.
- Figure 2,3,4 is not clear.

Experimental design

The research is within the Scope of the journal. The authors have addressed the reviewers' comments and concerns in their revised manuscript.

Validity of the findings

It should be emphasized that a Qualitative comparison with other architectures is made in Table 2.

Reviewer 7 ·

Basic reporting

This paper proposes a privacy protection method for smart city residents’ travel trajectories based on federal blockchain and interstellar file system, which assists government departments in effectively controlling the spread of epidemics while ensuring the security of city residents’ trajectory data.
Overall, this work clearly expresses the goals and contributions, especially the introduction and related work are well written. There is a strong correlation between the references and the research object of the paper. In the method section, the author provides a detailed introduction to the framework and implementation process of the proposed method. However, there are still some shortcomings in the details of this article that can be improved. For example, the terminology of the federated blockchain is not uniform enough. Some places use federated blockchain, while others use federal blockchain.

Experimental design

The experimental design involves using Hyperledger Fabric, a federated blockchain framework, combined with IPFS storage to achieve efficient data processing and protect personal privacy-related information. The method stores the ciphertext of complete personal travel trajectory data in decentralized IPFS storage, ensuring the security of stored data.
It is commendable that the author provided us with raw data and code. However, in the code address of Github, there is a lack of detailed explanation in the readme section. This is not conducive to other researchers reproducing the proposed method.

Validity of the findings

In the results and analysis section, the author verified the feasibility of the proposed solution through security analysis and performance evaluation. In the discussion section, a qualitative comparison was made between the proposed method and existing methods. However, it would be better to compare the experimental design from a quantitative perspective.

Reviewer 8 ·

Basic reporting

Paper is ok in terms of writing. However novelty is not good. Can you please provide experimental code ? I prefer to verify graphs(figure) from 7 to 11. They seems to be strange for me.

Also, you took some concepts from different papers without citing them. Try to cite this papers as well.
https://onlinelibrary.wiley.com/doi/epdf/10.1002/ett.4621

Experimental design

Experiment design is hard to check because I could not find any code to verify them.

Validity of the findings

We can only check with the code used to perform experiment. Authors can upload it on github and share link.

---

## Round 0.4 · Minor Revisions

Most of the comments have been addressed. The reviewers ask minor revisions.

Reviewer 6 ·

Basic reporting

The authors have addressed the reviewers' comments and concerns in their revised manuscript.
However, the references are still insufficient; in the section titled "IPFS in Data Privacy Protection" in line 159, current studies on privacy protection methods can be added.

Experimental design

In this method, the raw data is encrypted, and technical infrastructure is provided before it is sent to the IPFS storage and blockchain.

Validity of the findings

The proposed solution's performance evaluation and security analysis have been provided, and the comparison with other architectures has been made qualitatively.

Reviewer 7 ·

Basic reporting

In this paper, the authors propose a privacy protection method for smart city residents’ travel trajectories based on federated blockchain and interstellar file system, which assists government departments in effectively controlling the spread of epidemics while ensuring the security of city residents’ trajectory data. Overall, this work clearly expresses its goals and contributions, and has a certain degree of novelty. The authors have addressed the reviewers' comments and concerns in their revised manuscript. The structure conforms to PeerJ standards.

Experimental design

Methods described with sufficient detail & information to replicate. The author has provided raw data and experimental code, as well as detailed readme files, which have met the standard requirements of the journal.

Validity of the findings

Conclusions are well stated, linked to original research question & limited to supporting results. The author verified the feasibility of the proposed solution through security analysis and performance evaluation.

Reviewer 8 ·

Basic reporting

Authors updated the paper as per my comments. No further update needed from my side.

Experimental design

Perfect

Validity of the findings

Perfect

Additional comments

Perfect

---

## Round 0.5 · Major Revisions

The referral process is now complete. The referees and I feel that more work should be done. My decision is therefore to provisionally accept your paper subject to major revisions. This does not mean that your paper will be accepted after the revision.

Reviewer 6 ·

Basic reporting

The authors have addressed the reviewers' comments and concerns in their revised manuscript.

Experimental design

The article has a technical infrastructure that protects data privacy and is within the Aims and Scope of the journal.

Validity of the findings

In the proposed architecture, the results are related to and supportive of the study subject of the article.

Reviewer 8 ·

Basic reporting

The research paper titled "A novel privacy protection method of residents' travel trajectories based on federated blockchain and InterPlanetary file systems in smart cities" proposes an interesting approach to address privacy concerns in smart cities. However, there are several areas that can be improved to enhance the quality and effectiveness of the research:

Clarity of the Problem Statement: The paper should provide a clear and concise explanation of the specific privacy concerns associated with residents' travel trajectories in smart cities. It should outline the existing challenges and limitations of the current methods used for privacy protection. By doing so, readers can better understand the context and importance of the proposed method.

Literature Review: The research paper should include some recent comprehensive literature review that covers relevant studies, existing techniques, and approaches related to privacy protection in smart cities. This would help establish the novelty and contribution of the proposed method in comparison to existing solutions. e.g. A Privacy Preserving Internet of Things Smart Healthcare Financial System, IoTJ

Methodological Rigor: The paper should provide a detailed and well-defined methodology for implementing the proposed method. This includes describing the specific components of the federated blockchain and InterPlanetary file systems, as well as how they are integrated to protect residents' travel trajectories. The research should also consider potential limitations, challenges, and trade-offs of the proposed approach.

Discussion and Analysis: The paper should include a comprehensive discussion of the results obtained from the evaluation. This should include an analysis of the strengths and weaknesses of the proposed method, potential vulnerabilities or limitations, and suggestions for further improvements. It is important to address any potential ethical or security concerns that may arise from implementing the proposed method in real-world scenarios.

Conclusion and Future Work: The research paper should provide a well-summarized conclusion that highlights the main contributions of the study. Additionally, it should identify potential directions for future research and improvements in the proposed method. This could include exploring alternative technologies, addressing scalability issues, or considering additional privacy protection mechanisms.

By addressing these areas for improvement, the research paper can enhance its overall quality, readability, and scientific rigor, making it more impactful and valuable to the research community working on privacy protection in smart cities.

Experimental design

To validate the effectiveness of the proposed method, the research should conduct experiments or simulations using real-world or realistic data sets. The paper should provide a clear description of the evaluation metrics used, the experimental setup, and the results obtained. Comparisons with existing methods or benchmarks would further strengthen the validity of the proposed approach.

Validity of the findings

see basic reporting

---

## Round 0.6 · accepted · Accept

We are happy to inform you that your manuscript has been accepted since all of the reviewers are happy with the revisions.

Reviewer 8 ·

Basic reporting

Authors updated the paper as per my previous comments.

Experimental design

As above

Validity of the findings

As above